

# Minimal CMIP Emulator (MCE v1.2): A new simplified method for probabilistic climate projections

Junichi Tsutsui[1]

[1]Environmental Science Research Laboratory, Central Research Institute of Electric Power Industry, Abiko, 270-1194, Japan

*Correspondence to*: Junichi Tsutsui (tsutsui@criepi.denken.or.jp)

**Abstract.** Climate model emulators have a crucial role in assessing warming levels of many emission scenarios from probabilistic climate projections, based on new insights into Earth system response to $CO_2$ and other forcing factors. This article describes one such tool, MCE, from model formulation to application examples associated with a recent model intercomparison study. The MCE is based on impulse response functions and parameterized physics of effective radiative

forcing and carbon uptake over ocean and land. Perturbed model parameters for probabilistic projections are generated from statistical models and constrained with a Metropolis-Hastings independence sampler. A part of the model parameters associated with $CO_2$-induced warming have a covariance structure, as diagnosed from complex climate models of the Coupled Model Intercomparison Project (CMIP). Although perturbed ensembles can cover the diversity of CMIP models effectively, they need to be constrained toward substantially lower climate sensitivity for the resulting historical warming to

agree with the observed trends over recent decades. The model's simplicity and resulting successful calibration imply that a method with less complicated structures and fewer control parameters offers advantages when building reasonable perturbed ensembles in a transparent way. Experimental results for future scenarios show distinct differences between CMIP- and observation-consistent ensembles, suggesting that perturbed ensembles for scenario assessment need to be properly constrained with new insights into forced response over historical periods.

**1 Introduction**

Climate model emulators, or simple climate models, are numerical tools for representing the complex Earth system in reduced dimensions using aggregated variables, such as global mean surface temperature (GMST) and global $CO_2$ uptake over ocean and land. They offer advantages of ease and transparency, with a wide range of applications in both scientific and decision-making contexts (Schwarber et al., 2019). Their computational efficiency allows users to conduct climate

experiments for a number of emission scenarios with many different model parameters, to derive probabilistic climate projections. This article describes one such tool, Minimal CMIP Emulator (MCE), intended to emulate state-of-the-art atmosphere-ocean general circulation models (AOGCMs) in the Coupled Model Intercomparison Project (CMIP, Meehl et al., 2014) with sufficient simplicity and accuracy.



One key emulator application is climate assessment of emission scenarios presented in Intergovernmental Panel on Climate

Change (IPCC) reports. In the case of the 2014 Working Group III Fifth Assessment Report (AR5), over 1000 scenarios were assessed with a well-established emulator, MAGICC version 6 (Meinshausen et al., 2011), from its 600-member parameter ensemble runs (Clarke et al., 2014). The method used was based on Meinshausen et al. (2009) and has a range of future temperature increases similar to that of multiple AOGCMs from the CMIP Phase 5 (CMIP5, Taylor et al., 2012). The results from ensemble runs were used to classify each scenario by climate indicators associated with warming levels and to

probabilistically assess consistency with long-term temperature goals for climate change mitigation. The output of the CMIP5 models constitutes a dominant part of the scientific basis of AR5, and the specific emulator plays a crucial role in synthesizing the most comprehensive information on climate projections available at the time.

However, climate assessment of AR5 is regarded as indicative as it is based on a single probability distribution (Clarke et al., 2014). This is similar to the scenario assessment of the 2018 IPCC Special Report on global warming of 1.5 °C (SR15)

(Rogelj et al., 2018), where the same method as in AR5 was used for scenario classification but noticeable differences in radiative forcing and temperature response were identified between the results of MAGICC and of a different emulator, FaIR version 1.3 (Smith et al., 2018). FaIR incorporates recent updates of radiative forcing and shows greater non-$CO_2$ anthropogenic forcing in historical and future periods than MAGICC (Forster et al., 2018). In contrast, current and projected warming is generally greater in MAGICC than in FaIR, implying greater climate sensitivity in the former.

With regard to climate sensitivity, the new CMIP Phase 6 (CMIP6, Eyring et al., 2016) has been providing different outcomes from CMIP5. Equilibrium climate sensitivity (ECS), a hypothetical value of global warming at equilibrium in response to a doubling of the atmospheric $CO_2$ concentration, is generally greater in CMIP6 models than in CMIP5 models, mainly attributed to the models' cloud processes (Zelinka et al., 2020). Transient climate response (TCR), a value of global warming at the time of $CO_2$ doubling with an idealized 1%-per-year concentration increase, is also greater in CMIP6 than in

CMIP5 models, but their relative difference is reduced from that of ECS (Meehl et al., 2020). This characteristic, reflecting the tendency of realized warming fractions, is consistent with a theoretical relationship between climate feedback strength and thermal response timescales (Tsutsui, 2020). However, modeled historical warming generally appears greater in the CMIP5 models than in the CMIP6 models, suggesting that aerosol cooling is extremely strong in several CMIP6 models (Flynn and Mauritsen, 2020).

These confusing results require a more advanced and transparent methodology to synthesize new insights into forcing, response, and sensitivity, not only from climate modeling but also from other lines of evidence. The Reduced Complexity Model Intercomparison Project (RCMIP, Nicholls et al., 2020) is promising, providing the first comprehensive model intercomparison of emulators. During Phase 1 of this project, a new framework was established to systematically evaluate multiple emulators from scenario experiments that mirror those in CMIP5 and CMIP6, and 15 emulators were compared in

terms of their ability to approximate each of the CMIP6 models, mainly in terms of global mean temperature changes (Nicholls et al., 2020). Phase 2 then focused on probabilistic climate projections and nine models were compared under the same set of constraints for model parameter perturbations (Nicholls et al., submitted).



The MCE has been used in both phases, and the present article provides details of the version used in Phase 2. The MCE model consists of prediction equations for thermal response and carbon cycle. Although there are many emulators with different complexities, their core modules appear to be based on a few pioneering works and are often shared between different emulators. The thermal response of the MCE is implemented as a pure impulse response model (IRM), which is the most simplified form originated from the one presented in Hasselmann et al. (1993). The carbon cycle of the MCE is based on a part of the nonlinear impulse-response model of the coupled carbon-climate system (NICCS, Hooss et al., 2001), which may be categorized to be of intermediate complexity among RCMIP participants. One of them, ACC2 (Tanaka et al., 2007), also adopts the NICCS-based carbon cycle.

Although complex formulations generally are more capable of emulation, they are not necessarily advantageous for emulating individual CMIP models and representing their uncertainty ranges. For thermal response, this has been confirmed by the author's previous studies (Tsutsui, 2017; Tsutsui, 2020), which have demonstrated that a simple IRM can accurately emulate a variety of CMIP models in terms of temperature response to $CO_2$ forcing and provide a basis of parameter sampling that covers model diversity. These findings also imply that less complex emulators are suitable for knowledge transfer in a transparent way. From this perspective, key considerations for emulator design are in its subsidiary components, such as forcing parameterizations, treatment of non-linear processes involving some state-dependent response properties, and parameter constraining for probabilistic projections.

The remainder of this article is structured as follows. Section 2 describes model formulations and parameter sampling methods. Section 3 presents the experimental application of probabilistic climate projections. Section 4 discusses emulator performance and constraining model parameters. Finally, Section 5 presents the study's main conclusions.

## 2 Model description

### 2.1 Impulse response models

The MCE model is essentially built on impulse response functions for the fraction of the total $CO_2$ emitted that remains in the atmosphere (termed the airborne fraction), the decay of land carbon accumulated by the $CO_2$ fertilization effect, and temperature change to radiative forcing of $CO_2$ and other forcing agents. Under the linear response assumption with regard to input forcing $F$, an impulse response model (IRM) expresses the time change of a response variable $x$ by a convolution integral:

$$x(t) = \int_0^t F(t') \sum_i A_i \exp\left(-\frac{t-t'}{\tau_i}\right) dt', \tag{1}$$

where $t$ is time, and the sum of exponentials is an impulse response function with parameters $A_i$ and $\tau_i$ denoting the $i$-th component of the response amplitude and time constant, respectively. The time derivative of this equation is given by:



$$\frac{dx(t)}{dt} = \sum_i \left[ F(t)A_i - \frac{x_i(t)}{\tau_i} \right], \qquad (2)$$

or an equivalent box model form that is converted into the original IRM through Laplace transform or eigenfunction expansion. The time derivative implemented in the MCE uses an IRM form for land carbon decay and temperature change,

and a box model form for the airborne fraction, to address partitioning of excess carbon between the atmosphere and ocean mixed layer.

The IRM for the airborne fraction defines five components, one of which has infinity time constant, paired with an amplitude corresponding to an asymptotic long-term fraction. In the current configuration, the remaining four time constants are fixed at 236.5, 59.52, 12.17, and 1.271 years, adjusted to a specific three-dimensional ocean carbon cycle model in Hooss et

al. (2001). The corresponding amplitudes assume perturbations at reference values of 0.24, 0.21, 0.25, and 0.1, respectively, with a reference long-term airborne fraction of 0.20. These reference values and perturbation ranges are set empirically so that resulting carbon budgets—cumulative land and ocean $CO_2$ uptake—agree with those of historical observations and CMIP experiments.

As described below, IRM parameters are converted into a set of parameters for an equivalent box model dealing with carbon

exchange between four layers. In this conversion, the response of the shortest timescale component is interpreted as equilibration between the atmosphere and ocean mixed layer, which are combined into a composite layer in the box model. Figure 1 shows response to an idealized instantaneous input of 100 GtC without land carbon uptake and climate feedback. In this case, the airborne fraction decreases from 0.9 to a long-term equilibrium of about 0.2 at a gradually decreasing rate.

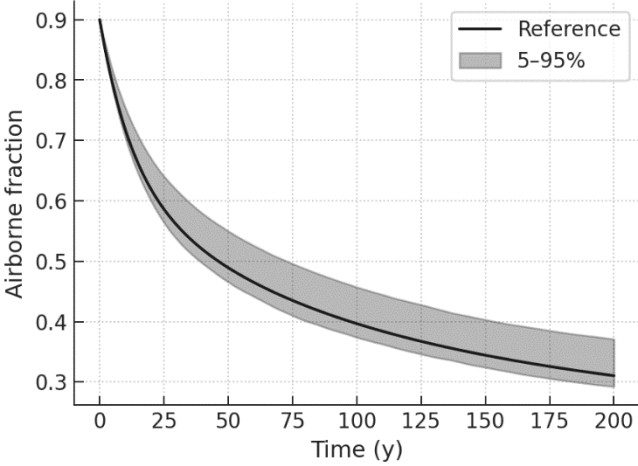

**Figure 1: Response of the airborne fraction to an initial input of 100 GtC without land $CO_2$ uptake and climate feedback. The line shows the case of reference amplitudes, and shading shows the range of the 5th–95th percentiles of the 'Prior' ensemble, described in 3.1.**

The IRM for land carbon defines four carbon pools, representing ground vegetation, wood, detritus, and soil organic carbon, with distinct overturning times ($\tau_i$). The forcing term ($F$) is net primary production (NPP) enhanced by the effect of $CO_2$

fertilization, generally expressed by $\beta_L N_0$, where $\beta_L$ is a fertilization factor that depends on the atmospheric $CO_2$





concentration, and $N_0$ is base annual NPP in GtC per year. The response amplitude ($A_i$) is rewritten as $\tilde{A}_{bi}\tau_i$, where $\tilde{A}_{bi}$ denotes a decay flux after an initial carbon input. Based on Joos et al. (1996), the IRM parameters of the four carbon pools are set to 2.9, 20, 2.2, and 100 years for $\tau_i$, and 0.70211, 0.013414, −0.71846, and 0.0029323 years$^{-1}$ for $\tilde{A}_{bi}$, respectively. Figure 2 illustrates response to unit forcing in this configuration.

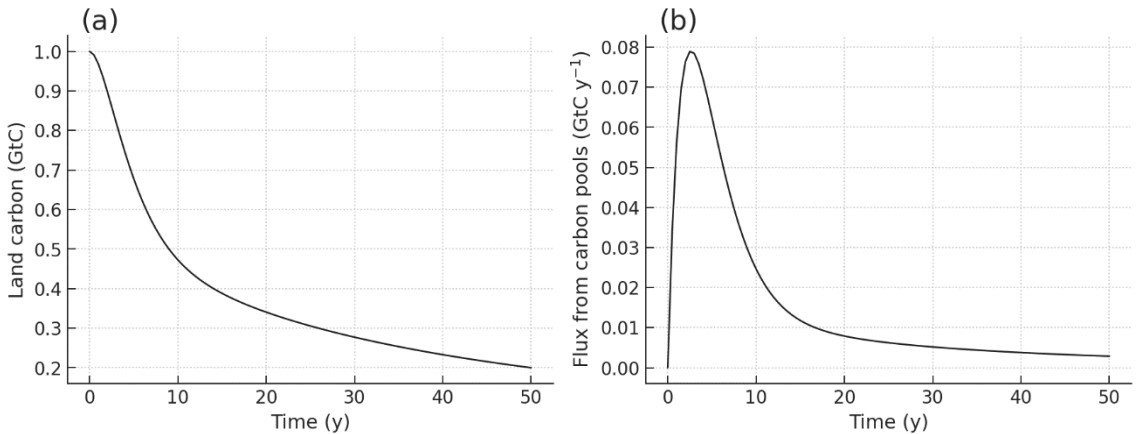

**Figure 2: Response of land carbon to instantaneous unit input (a), and accompanying flux from carbon pools (b).**

In addition, the MCE deals with temperature dependency for the time constants of wood and soil organic carbon, indicating the tendency for warming to accelerate the decomposition of organic matter. This is one of the climate-carbon cycle feedback processes and is implemented with an adjustment coefficient varied along a logistic curve with respect to surface warming, as illustrated in Fig. 3. This scheme has a parameter to control the asymptotic minimum value of the coefficient. The figure shows three curves with different control parameters, corresponding to the 17$^{th}$, 50$^{th}$, and 83$^{rd}$ percentiles of the 'Prior' ensemble, described in 3.1, adjusted to be consistent with the variation of CMIP Earth system models (ESMs). In the IRM form, land accumulated carbon is proportional to $\sum_i \tilde{A}_{bi}\tau_i^2$, expressing the remaining carbon at an equilibrium state under unit continuous input, and the decrease in the time constants affects accumulated carbon quadratically.

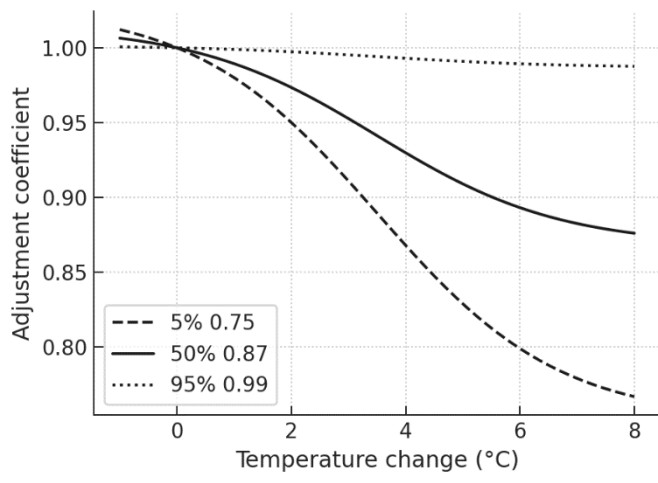



**Figure 3: Adjustment coefficient as a function of surface temperature change to multiply the time constants for the decay of wood and soil organic matter. The three curves are functions corresponding to the 5th, 50th, and 95th percentiles of the 'Prior' ensemble, described in 3.1, with different asymptotic minimum values, as described in the legend. The temperature at which a curve has the maximum gradient is fixed at 3.5 °C.**

The IRM of the temperature change defines three components with typical time constants of approximately 1, 10, and > 100 years. Although the temperature response is usually well represented by two separated time constants of approximately 4 and > 100 years (e.g., Held et al., 2010, Geoffroy et al., 2013), dividing fast response is advantageous when considering instantaneous forcing changes, such as volcanic eruptions and geoengineering mitigation. The response amplitude is rewritten by $\tilde{A}_i/(\lambda\tau_i)$, where $\tilde{A}_i$ is normalized so that the component sum is unity, and $\lambda$ is the climate feedback parameter in

$Wm^{-2}°C^{-1}$, defined as the derivative of the outgoing thermal flux with respect to temperature change. These IRM parameters can be adjusted to emulate individual CMIP models with sufficient accuracy, as demonstrated in Tsutsui (2020), which serves a basis to build a perturbed parameter ensemble.

## 2.2 Carbon uptake over ocean

The box model converted from the IRM for the airborne fraction is as follows:

$$\frac{dc_0}{dt} = -\frac{\eta_1}{h_s}c_s + \frac{\eta_1}{h_1}c_1 + e - f, \tag{3}$$

$$\frac{dc_1}{dt} = \frac{\eta_1}{h_s}c_s - \frac{\eta_1 + \eta_2}{h_1}c_1 + \frac{\eta_2}{h_2}c_2, \tag{4}$$

$$\frac{dc_2}{dt} = \frac{\eta_2}{h_1}c_1 - \frac{\eta_2 + \eta_3}{h_2}c_2 + \frac{\eta_3}{h_3}c_3, \tag{5}$$

$$\frac{dc_3}{dt} = \frac{\eta_3}{h_2}c_2 - \frac{\eta_3}{h_3}c_3, \tag{6}$$

where $c_k$ is the amount of excess carbon in layer $k$, $h_k$ is the layer depth, $\eta_k$ is the exchange coefficient between layer $k-1$

and layer $k$, $e$ is anthropogenic emissions, and $f$ is natural uptake over land. The parameters $h_k$ and $\eta_k$ are set through numerical optimization for the box model to be equivalent to the IRM form. The top layer, indexed with "0," is the composite atmosphere-ocean mixed layer, and the three sub-surface layers are indexed with "1," "2," and "3" in the order of ocean depth. The amount of excess carbon in the top layer ($c_0$) is partitioned into atmospheric and ocean components, denoted by $c_a$ and $c_s$, subject to chemical equilibrium at the ocean surface. The carbon exchange between the top layer and

the first sub-surface is expressed in terms of $c_s$.

For a given time series of $CO_2$ emissions (emission-driven) or atmospheric $CO_2$ concentrations (concentration-driven), time integration is performed. In the latter case, $c_0$ and its partition within the composite layer are diagnostically determined, and the top-layer equation is dropped.





The partition of $c_0$ is solved through numerical computation with regard to hydrogen ion concentration associated with

changes in dissolved inorganic carbon concentration (DIC) under the assumption of constant alkalinity (Alk). DIC, defined

as the sum of $[CO_2]$, $[HCO_3^-]$ and $[CO_3^{2-}]$, where $[x]$ denotes the concentration of a substance $x$ in mol $L^{-1}$, is expressed as:

$$DIC = \left(1 + \frac{K_1}{[H^+]} + \frac{K_1 K_2}{[H^+]^2}\right)[CO_2], \qquad (7)$$

where $K_1$ and $K_2$ are equilibrium constants for bicarbonate and carbonate, defined as $K_1 = [H^+][HCO_3^-]/[CO_2]$ and $K_2 = [H^+][CO_3^{2-}]/[HCO_3^-]$. $[CO_2]$, defined as the sum of $[CO_2(aq)]$ and $[H_2CO_3(aq)]$ is related to the partial pressure of $CO_2$

($pCO_2$) with equilibrium constant $K_0$, as in $K_0 = [CO_2]/pCO_2$. Alkalinity, here considering borate ions as well as

bicarbonate and carbonate ions, is represented as:

$$Alk = \frac{K_1[H^+] + 2K_1 K_2}{[H^+]^2}[CO_2] + \frac{K_b B_T}{K_b + [H^+]} + \frac{K_w}{[H^+]} - [H^+], \qquad (8)$$

where $B_T$ is total borate concentration $[B(OH)_3] + [B(OH)_4^-]$, and $K_b$ and $K_w$ are equilibrium constants for borate and water,

defined as $[H^+][B(OH)_4^-]/[B(OH)_3]$ and $[H^+][OH^-]$.

The values of Alk, $B_T$, and the equilibrium constants of $K_0$, $K_1$, $K_2$, $K_b$, and $K_w$ are set based on Dickson et al. (2007). The

equilibrium constants depend on water temperature, and carbon uptake decreases with temperature, representing a climate-

carbon cycle feedback process. This temperature dependency is implemented as a linear regression for empirical expressions,

as shown in Fig. 4.

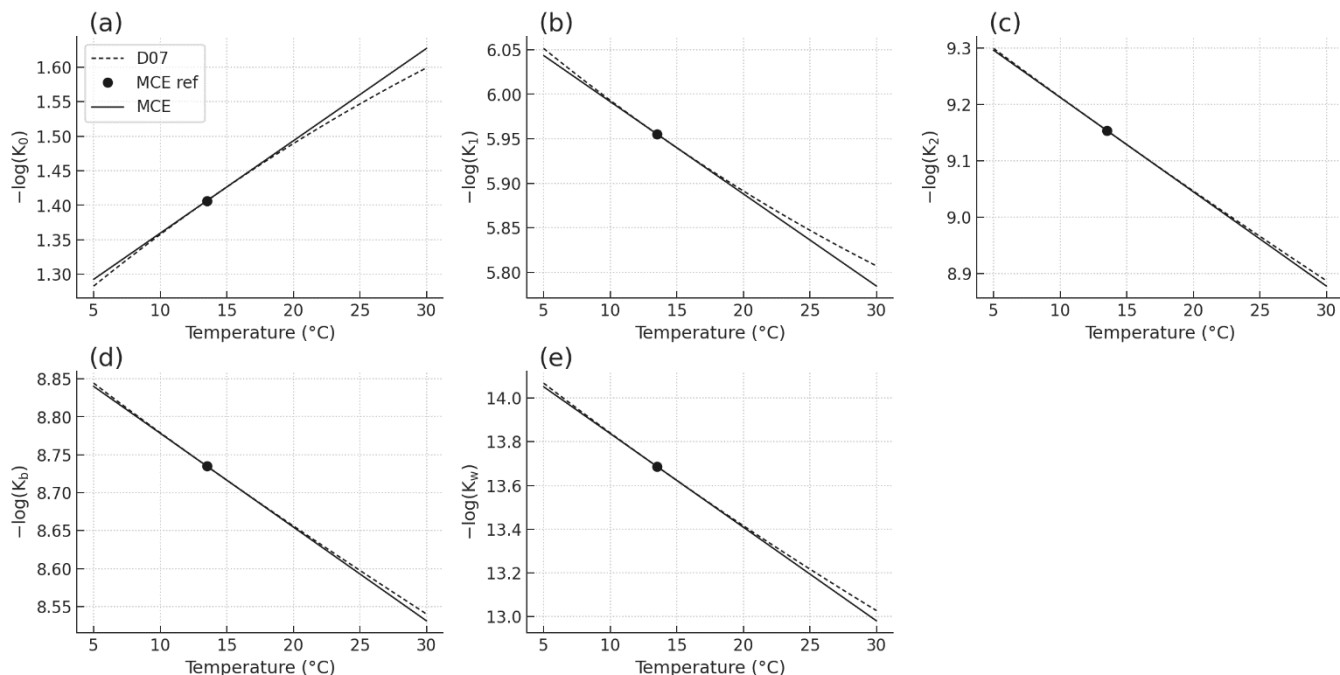

**Figure 4: Temperature-dependent equilibrium constants of $K_0$, $K_1$, $K_2$, $K_b$, and $K_w$ (a–e) in the MCE model (solid lines), which approximate empirical expressions in Dickson et al. (2007) (D07, dotted lines). Values at a reference seawater temperature of 13.5 °C (dots) are assigned to those in the MCE's preindustrial state.**



The amount of excess carbon that can be accumulated in the ocean is proportional to a change in DIC from its preindustrial value. This carbon uptake potential and its temperature dependency are illustrated in Fig. 5. $CO_2$-induced global warming

increases the airborne fraction in two ways—through the buffering mechanism of seawater and through temperature dependency of chemical equilibrium. The former is shown as a decreasing change rate of the DIC with regard to atmospheric $CO_2$ concentration (Fig. 5(a)), while the latter is shown as reduction in rates of carbon uptake potential with temperature, which also depends on the concentration (Fig. 5(b)). The reduction rate is approximately proportional to the warming level, typically about 4 % per 1 °C at doubling $CO_2$.

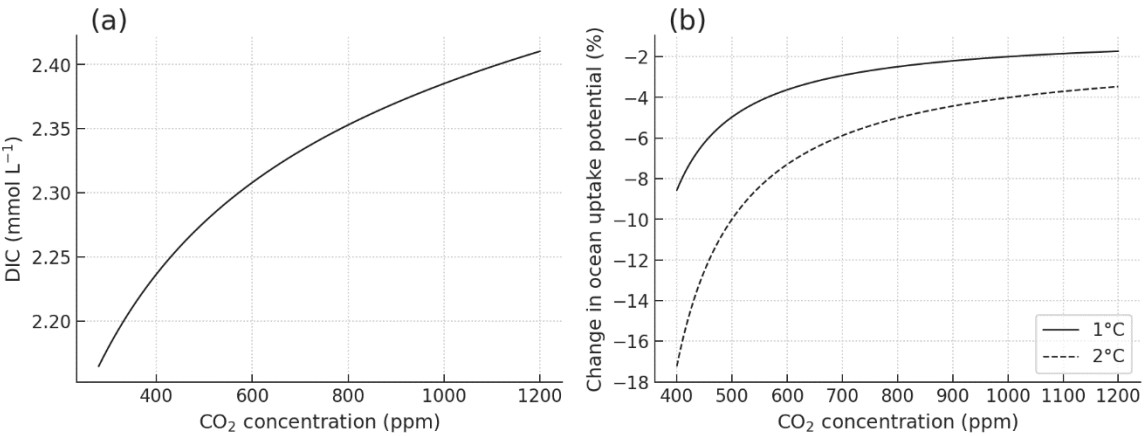


**Figure 5: DIC in the ocean mixed layer as a function of atmospheric $CO_2$ concentration (a) and changes in ocean carbon uptake potential, measured by increase in DIC from preindustrial levels, due to 1 °C and 2 °C warming (b). The preindustrial $CO_2$ concentration is assumed to be 284.317 ppm and preindustrial DIC is about 2.17 mmol L$^{-1}$.**

### 2.3 CO₂ fertilization

The land carbon uptake term $f$ in Eq. (3) is calculated from Eq. (2), rewritten as:

$$f(t) = \sum_i \left[ \beta_L(t) N_0 \tilde{A}_{bi} \tau_i - \frac{c_{bi}}{\tau_i} \right], \tag{9}$$

where $c_{bi}$ is the $i$-th component of accumulated carbon by $CO_2$ fertilization. The base NPP ($N_0$) is set to 60 GtC/y and the fertilization factor ($\beta_L$) is formulated with a sigmoid curve with regard to $CO_2$ concentration $C(t)$, as described in Meinshausen et al. (2011). This implementation is connected to a conventional logarithmic formula:

$$\beta_L = 1 + \hat{\beta}_L \ln\left[ \frac{C(t)}{C(0)} \right], \tag{10}$$

such that the sigmoid and logarithmic curves are equal in terms of an increase ratio at 680 ppm relative to 340 ppm, and the latter factor $\hat{\beta}_L$ is used as a control parameter. Figure 6 illustrates three curves with different control parameters in the MCE model.





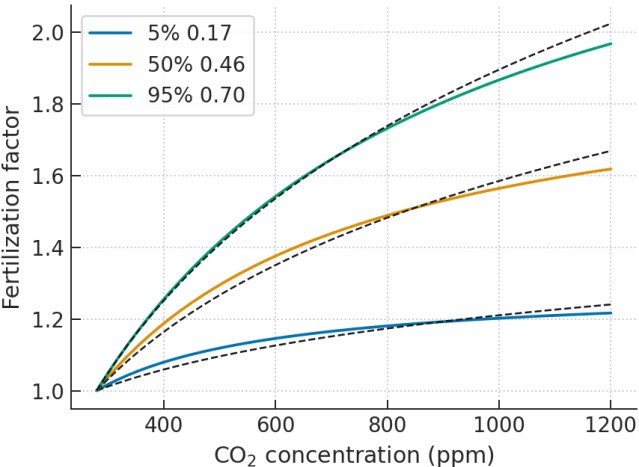

**Figure 6: $CO_2$ fertilization factor as a function of atmospheric $CO_2$ concentration with different control parameters ($\hat{\beta}_L$) of 0.17, 0.45, and 0.69, corresponding to the 5th, 50th, and 95th percentiles of the 'Prior' ensemble described in 3.1. The colored lines show sigmoid curves used in the MCE model and the black dashed lines show reference logarithmic curves.**

## 2.4 Effective radiative forcing

The forcing term in the IRM for temperature change is assumed to be effective radiative forcing (ERF), defined as top-of-atmosphere (TOA) radiative imbalance due to a change in a forcing agent through rapid adjustments in the stratosphere and troposphere prior to a response in surface temperature (Myhre et al., 2013, Sherwood et al., 2015). Forcing, defined as such, serves as a good predictor of surface temperature change.

$CO_2$ forcing is evaluated with the following quadratic formula, in terms of the logarithm of $CO_2$ concentration:

$$F_C(x) = (\beta_C - 1)[\hat{F}_C(x) - 2F_C(2)]\left[\frac{2\hat{F}_C(x)}{F_C(2)} - 1\right] + \beta_C\hat{F}_C(x) \qquad (11)$$

$$\hat{F}_C(x) = \alpha_C\ln\left[\frac{CO_2(t)}{CO_2(0)}\right], \qquad (12)$$

where $x$ is the ratio of $CO_2$ concentrations to a preindustrial level, $\alpha_C$ is a scaling parameter in $Wm^{-2}$, and $\beta_C$ is an amplification factor defined as $F_C(4) = \beta_C \times \hat{F}_C(4)$. This scheme was presented in Tsutsui (2017) to emulate the CMIP's core $CO_2$ increase experiments for instantaneous quadrupling and 1%-per-year increase, referred to as abrupt-4xCO2 and 1pctCO2, respectively. The two control parameters are diagnosed consistently with IRM parameters for individual CMIP models (Tsutsui 2020). The current diagnosing procedure solves numerical optimization to approximate the first 150-year and 140-year time series from abrupt-4xCO2 and 1pctCO2 experiments, respectively, in terms of TOA energy imbalance and the surface air temperature anomaly, respectively. The quadratic term is activated when the concentration exceeds a two-times level ($x > 2$), and $\beta_C$ is set to unity in the range $x \le 2$ so that $F_C$ is equivalent to $\hat{F}_C$. The forcing amplification is expected to be valid in the range $x \le 4$ and the quadratic term is dropped beyond a four-time level. Figure 7 illustrates example outputs of the $CO_2$ forcing scheme in a range of 5th–95th percentiles of the 'Prior' ensemble for control parameters.



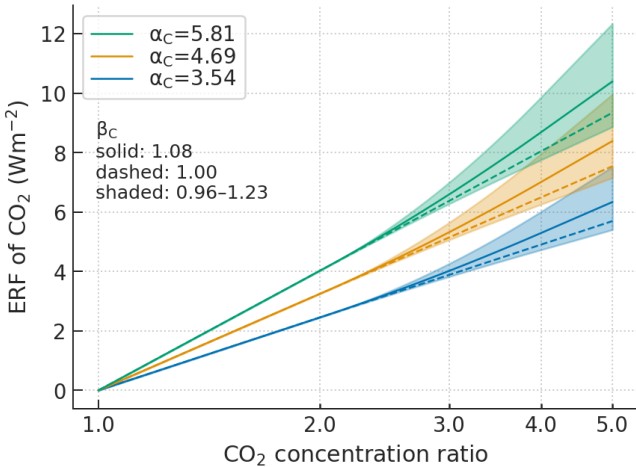

**Figure 7: Effective radiative forcing (ERF) of CO₂ as a function of the ratio of CO₂ concentrations to a preindustrial level. The scaling parameter $\alpha_C$ is set to three different values, corresponding to the 5[th], 50[th], and 95[th] percentiles of the 'Prior' ensemble, described in 3.1. For each $\alpha_C$ value, the amplification factor $\beta_C$ is varied between the 5[th] and 95[th] percentiles (shaded area), and is set to two specific values of the 50[th] percentile (solid line) and unity (dashed line, no amplification).**

The forcing of CH₄ and N₂O is evaluated with the expressions given in Etminan et al. (2016). The forcing of halogenated gases is simply calculated as changes in concentration from preindustrial levels multiplied by radiative efficiencies assessed in the latest IPCC report (currently AR5, Myhre et al., 2013).

The current MCE model does not support non-CO₂ gas cycles and ERF schemes for other forcing agents, such as aerosols, tropospheric and stratospheric ozone, solar radiation, and volcanic eruptions. Experiments considering non-CO₂ forcing require prescribed concentrations for long-lived greenhouse gases (GHGs) and prescribed ERF for others.

**2.5 Parameter sampling**

Probabilistic runs use an ensemble of perturbed model parameters designed to encompass the variation of multiple CMIP models with additional constraints with regard to assessed ranges of key climate indicators. In general, a series of candidate values of an uncertain parameter is generated from its statistical model and, if necessary, sampled from the series with an acceptance algorithm for given constraints. The latter process is Bayesian updating from a prior probability distribution to a posterior and here uses a Metropolis-Hastings (MH) independence sampler. As mentioned above, uncertain parameters include IRM amplitudes for the airborne fraction, control parameters for land carbon decay timescales and CO₂ fertilization, IRM parameters for temperature change, and control parameters for the CO₂ forcing scheme.

The carbon cycle parameters are individually generated from a uniform distribution with a given mean and perturbation range. The means and ranges are determined on a trial basis so that ranges of carbon budgets in historical and scenario experiments are consistent with those from multiple CMIP ESMs. Since the sum of IRM amplitudes for the airborne fraction is unity, their perturbed values are normalized as such, subject to a modified distribution with more samples about the mean resulting from the operation.





The temperature response and $CO_2$ forcing parameters are synthetically generated from a multivariate normal distribution reflecting the variation of multiple CMIP AOGCMs. The IRM for temperature change has three pairs of time constant ($\tau_i$) and dimensional amplitude ($A_i$), and the $CO_2$ forcing scheme has two control parameters ($\alpha_C$ and $\beta_C$). A total of eight parameters have been diagnosed for each of the multiple CMIP models, revealing characteristic covariance structures, such as a noticeable negative correlation between feedback strength ($1/\lambda$) and a realized warming fraction (typically TCR-to-ECS

ratio), and a weakly negative correlation between the forcing scale ($\alpha_C$) and feedback strength (Tsutsui, 2020). The multivariate normal distribution is built on principal components (PCs) of these diagnosed parameters, as described in Tsutsui (2017).

The eight parameters to be fed into PC analysis can include some derived parameters from the following expressions:

$$A_i = \frac{\tilde{A}_i}{\lambda \tau_i}, \quad \sum_i \tilde{A}_i = 1, \tag{13}$$


$$\text{ECS} = \frac{\alpha_C \ln(2)}{\lambda}, \tag{14}$$

$$\text{ECS}_G = \frac{\alpha_C \beta_C \ln(2)}{\lambda}, \tag{15}$$

$$\text{TCR} = \text{ECS}\left\{1 - \sum_i \tilde{A}_i \frac{\tau_i}{t_{70}}\left[1 - \exp\left(-\frac{t_{70}}{\tau_i}\right)\right]\right\}, \tag{16}$$

where ECS is defined using a diagnosed forcing of $CO_2$ doubling, while $\text{ECS}_G$ uses $CO_2$ quadrupling with a factor of 0.5 as in Gregory et al. (2004). Eq. (16) is derived from time integration of Eq. (1) to the 70[th] year ($t_{70}$) along a 1 %-per-year

increasing path that defines TCR. One possible set consists of TCR, $\tilde{A}_0/\tilde{A}_2$, $\tilde{A}_1/\tilde{A}_2$, $\tau_0$, $\tau_1$, $\tau_2$, $\alpha_C$, and $\beta_C$, applied with logarithmic transformation, except for $\alpha_C$. This set was adopted in the experiments described below. The logarithmic transformation is intended to allow fair normality of PC scores, as a basis for fitting a multivariate normal distribution, and to make generated candidates positive.

Probabilistic runs can also use different scaling factors to adjust individual non-$CO_2$ ERF time series. This is a simple

implementation to deal with non-$CO_2$ forcing uncertainties, typically assessed as a range at a reference time point. The scaling factor is perturbed with a suitable statistical model fitted to the range.

All uncertain parameters and ERF scaling parameters are not necessarily independent. The current sampling procedure incorporates covariance between the eight parameters relevant to temperature change in response to $CO_2$ forcing. However, the procedure assumes no other correlations, implying that uncertainties of the $CO_2$-induced temperature response are

independent from those of the carbon cycle and non-$CO_2$ forcing.





## 3 Application examples

### 3.1 Scenario experiments

To demonstrate a typical application of the MCE model, a number of scenario experiments that mirror those of CMIP6 were conducted, including idealized abrupt-4xCO2 and 1pctCO2, and historical-future scenarios based on the Shared
Socioeconomic Pathways (SSPs, Riahi et al., 2017). In the latter experiments, the model was initialized for the year 1850 and driven with GHG concentrations and other prescribed ERF, both provided from the RCMIP (Nicholls et al., 2020).

For each scenario, two sets of 600-member ensemble runs were conducted; one was perturbed to be consistent with a CMIP multi-model ensemble and the other was further constrained according to the RCMIP Phase 2 protocol (Nicholls et al., submitted), here labeled 'Prior' and 'Constrained', respectively. 'Prior' refers to 25 CMIP5 and 38 CMIP6 AOGCMs for the
PC analysis input, and to 8 CMIP5 and 11 CMIP6 ESMs diagnosed in Arora et al. (2020) for simulated carbon budgets in the 1pctCO2 experiment. Diagnosed forcing/response parameters of the multiple AOGCMs are presented in the MCE's code repository.

The uncertain carbon cycle parameters for 'Prior' were generated from the above-mentioned statistical models, as shown in Figs. 1, 3, and 6, and were processed by the MH sampler to constrain accumulated land carbon at doubling $CO_2$ along the
1%-per-year pathway. In this case, 1pctCO2 scenario runs with a set of proposed parameters were conducted to obtain data fed into the sampler. This single constraint was selected as it works inclusively for other relevant constraints through a trade-off relationship between ocean and land in terms of accumulated carbon.

RCMIP Phase 2 defines a number of constraints for climate indicators, including ERF levels, carbon budgets, recent warming trends, and climate sensitivity metrics of ECS, TCR, and transient climate response to cumulative $CO_2$ emissions
(TCRE). Here, TCRE is defined as the ratio of the TCR to implied cumulative $CO_2$ emissions at the time of $CO_2$ doubling along the same 1 %-per-year trajectory as that for TCR. These constraints use literature-based assessed ranges, referred to as a "proxy assessment" to distinguish these from the formal IPCC assessment. The 'Constrained' uncertain parameters were sampled from those of 'Prior' through a sequence of the MH sampler with a subset of RCMIP constraints, as follows: (1) $CO_2$ ERF in 2014 relative to 1750 evaluated in Smith et al. (2020), (2) TCR estimated in Tokarska et al. (2020, Table S3,
both constrained), and (3) GMST in the period 1961–1990 relative to the period 2000–2019 from the HadCRUT.4.6.0.0 dataset (Morice et al., 2012) and ocean heat content (OHC) in 2018 relative to 1971 from the dataset of von Schuckmann et al. (2020). In this case, in addition to 1pctCO2 runs, historical scenario runs with a set of proposed parameters were conducted to obtain data fed into the sampler.

The IRM for temperature change is transformed into a three-layer heat exchange model in physical space. When diagnosing
the $CO_2$ forcing and response parameters, the top layer temperature was treated as global mean surface air temperature (GSAT). As in HadCRUT GMST was defined as a surface air ocean blended temperature change; here, a factor of 1.04 was used to convert observed GMST change into the MCE's GSAT change. Likewise, as the MCE's three layers cannot be





allocated to specific climate system components, a factor of 1.08 was used to convert observed OHC change into the MCE's total heat content change.

Besides $CO_2$ forcing, the RCMIP constraints include the ranges of non-$CO_2$ forcing over a historical period for $CH_4$, $N_2O$, halocarbons (aggregated into "Montreal gases" (CFCs/HCFCs/halons) and other "F-gases" (HFCs/PFCs/$SF_6$)), aerosols (aggregated), tropospheric ozone, stratospheric ozone, stratospheric water vapor from $CH_4$, and albedo change due to land use and black carbon aerosols on snow and ice. Ranges are based on AR5 (Myhre et al., 2013), except for those for $CH_4$ and aerosols, which consider recent updates (Etminan et al., 2016; Smith et al., 2020). To incorporate these uncertainty ranges in

historical-future scenarios, the scaling factors to adjust individual non-$CO_2$ ERF time series were perturbed using normal or skewed normal distributions fitted to the prescribed ranges.

The RCMIP constraints are provided as *likely* ranges and optionally *very likely* ranges, corresponding to 17–83% and 5–95% according to the IPCC's likelihood terms in italics. These ranges were applied to generate uncertain parameter proposals and to build the MH sampler requiring probability densities for a target distribution.

Other details of experimental specifications are provided in the MCE's code repository.

## 3.2 Results: climate indicators

Figure 8 illustrates relationships between key indicators associated with the carbon budget and climate sensitivity of the two ensembles in comparison with the CMIP models. The carbon budget is measured by the amount of accumulated carbon and its allocation to ocean and land reservoirs. Here, total accumulation and the ocean allocation ratio at doubling and

quadrupling $CO_2$ levels are used as key indicators. The CMIP ESMs indicate a clear negative correlation between the two quantities (Fig. 8 (a) and (b)), reflecting much greater uncertainties relating to land carbon. This feature is well represented by the MCE parameter ensembles. Although there are some model differences between CMIP5 and CMIP6 eras, such as a reduced model spread in the latter associated with nitrogen cycle implementation (Arora et al., 2020), the MCE ensembles currently do not distinguish between the two. The carbon indicators of the 'Constrained' ensemble do not differ significantly

from those of 'Prior' but are distributed toward higher total accumulations, attributed to warming differences that affect carbon cycle-climate feedbacks.



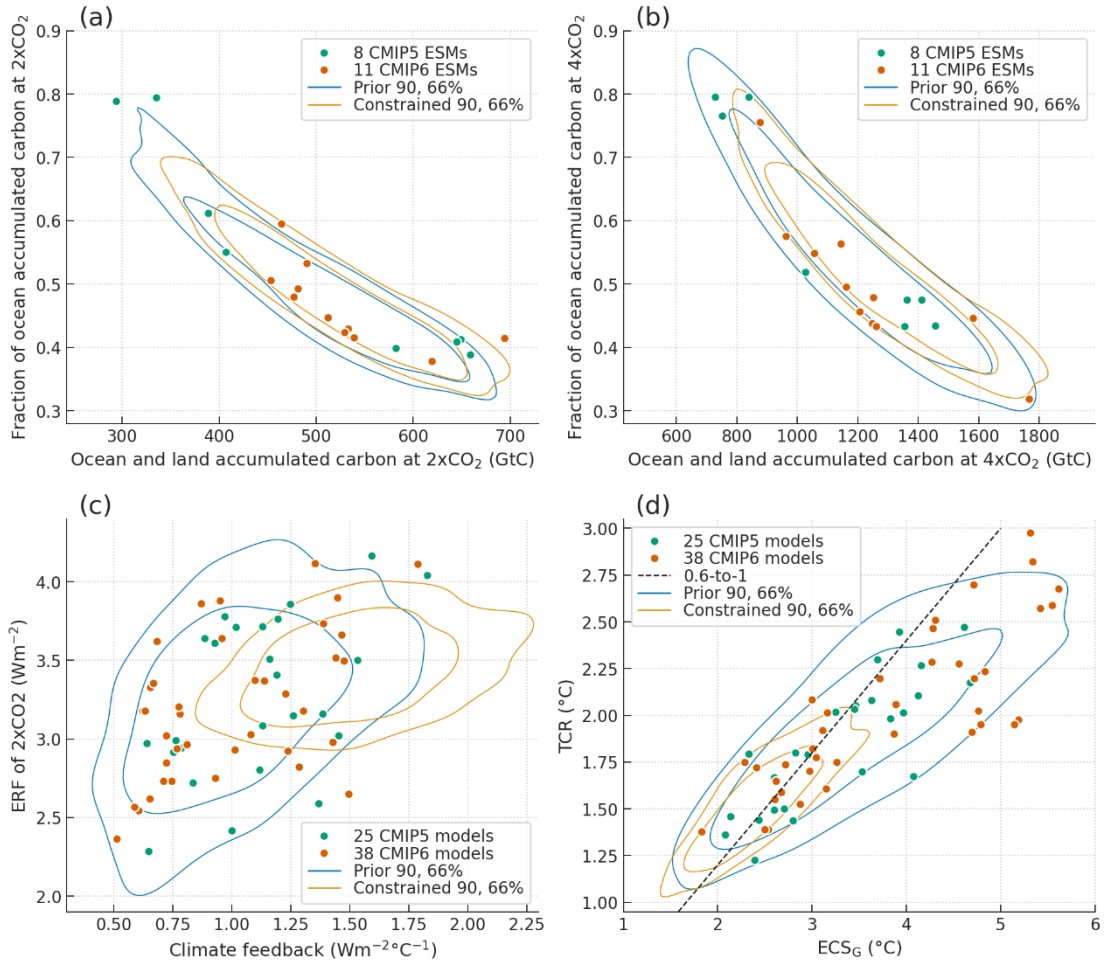

**Figure 8: Relationships between key indicators associated with carbon budget and climate sensitivity in comparison with CMIP**
**models. Contours indicate kernel density levels at which the circles cover 90% and 66% members. The legend indicates the**
**number of the CMIP models. (a) The fraction of ocean accumulated carbon and ocean and land totals in the 70th year of 1pctCO2.**
**(b) Same as panel (a), but in the 140th year. (c) Effective radiative forcing (ERF) of $CO_2$ doubling and climate feedback parameter.**
**(d) Transient climate response (TCR) and equilibrium climate sensitivity diagnosed from abrupt-4xCO2 ($ECS_G$). The dashed line**
**is located where the ratio of TCR to $ECS_G$ is 0.6 as a reference.**

In contrast, climate sensitivity differences are most prominent and well characterized with key indicators' distributions on
two-dimensional domains: the ERF of $CO_2$ doubling derived from $\alpha_C$ vs. the climate feedback parameter ($\lambda$) (Fig. 8 (c)), and
TCR vs $ECS_G$ (Fig. 8 (d)). While the 'Prior' distributions cover the CMIP AOGCMs effectively, the 'Constrained' are
confined to lower sensitivity values—greater $\lambda$ and smaller TCR and $ECS_G$, attributed to the observed GMST and OHC
constraints. The 'Prior' distribution of the $CO_2$ forcing agrees well with the CMIP distribution, which shows a weakly
positive correlation with the climate feedback parameter. In contrast, the 'Constrained' forcing levels are confined to an
upper half of the CMIP AOGCMs, attributed to the historical $CO_2$ forcing constraint, and the forcing-feedback correlation
becomes weak. Transient sensitivity is not necessarily proportional to equilibrium sensitivity, and greater CMIP6 sensitivity





is more evident in ECS$_G$ than in TCR. The inherent relationship between feedback strength and response timescales is responsible for the tendency, together with the forcing amplification effect represented by $\beta_C$. The PC analysis-based
statistical model captures such covariance structure effectively.

Figure 9 illustrates historical GMST and OHC of the MCE's two ensembles in comparison with their observations, from which constraining data are considered for recent warming trends. In the figure, the time series are adjusted relative to the reference period 1961–1990 for GMST and the reference year 1971 for OHC. While the 'Prior' series are well above the observed warming during the target period 2000–2019 for GMST and during the target year 2018 for OHC, the 'Constrained'
agree well with recent trends. The observation-based constraining results in lower climate sensitivity in the latter ensemble. However, considerable uncertainties remain with regard to longer trends and unforced climate variability. In an earlier period, observed GMST was rather close to 'Prior', and the 'Constrained' trend appears underestimated. The OHC trend cannot be validated owing to its limited observation period. Assessment of forced response in the historical period, which is currently not available, would allow more reliable parameter sampling.

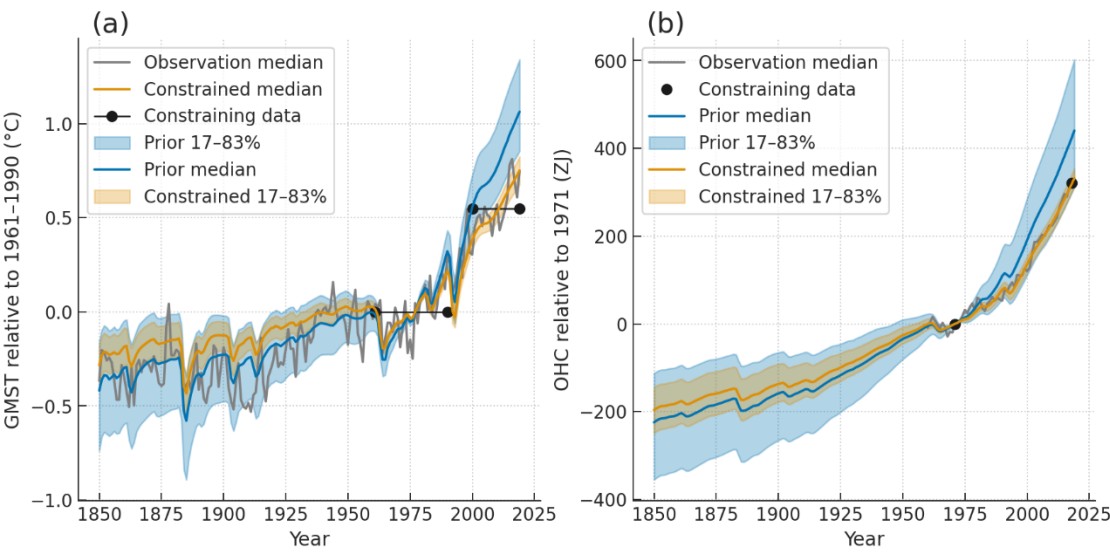


**Figure 9: (a) Historical global mean surface temperature (GMST) relative to 1961–1990 and (b) historical ocean heat content (OHC) relative to 1971 in the period 1850–2019 compared with observation data from HadCRUT4.6.0.0 for GMST and von Schuckmann et al. (2020) for OHC. The black dots indicate the levels at two different periods or years used for the observation constraints.**

The greater warming in 'Prior' is not only due to its greater climate sensitivity, but also partly due to non-CO$_2$ forcing differences, as shown in Fig. 10. The scaling factors of the non-CO$_2$ forcing agents are independently perturbed in the 'Prior' ensemble and probabilistically selected through the series of the MH sampler. Although the sampling process does not directly refer to forcing levels of non-CO$_2$ agents, it can modify their distributions to be consistent with other constraints. This modification is found for non-CO$_2$ GHGs and ozone time series (Fig. 10 (b)), and the most dominant contribution is of
Montreal-gases (not shown). The ERF of Montreal-gases rapidly increases from the 1960s and levels off from the 1990s, and the sampling results imply that this tendency is not consistent with the recent warming trend. Total ERF fluctuates with





changes in solar irradiance and volcanic eruptions, for which the RCMIP's prescribed forcing was used without their efficacy uncertainties.

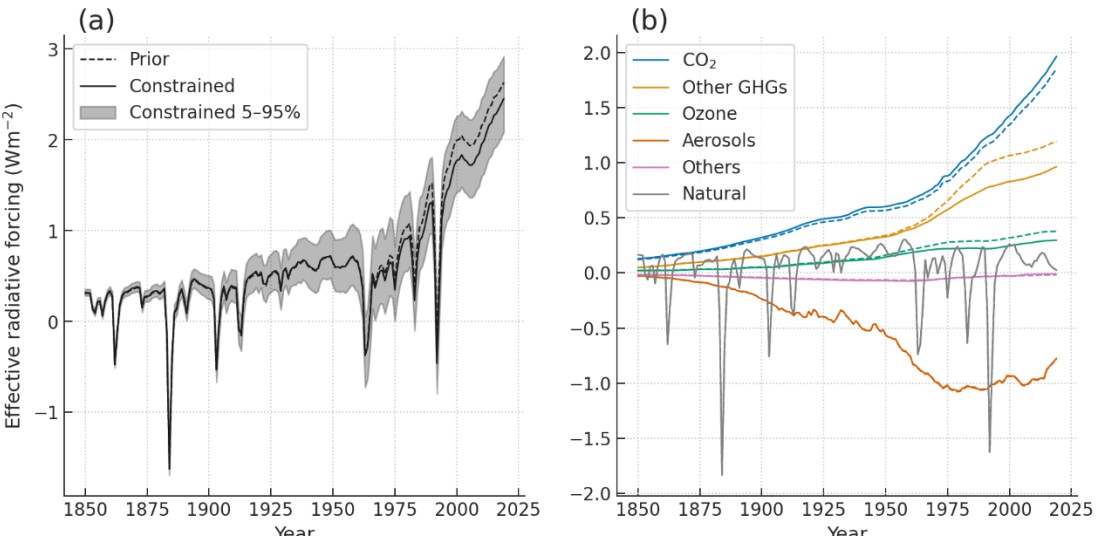

**Figure 10: Historical effective radiative forcing (ERF) in the period 1850–2019 for total ERF (a) and aggregated components (b). The ensembles' medians are shown by lines, and the 5–95% range of the 'Constrained' ensemble is shown for the total by shading.**

Figure 11 displays the ranges of climate indicators from the two ensembles associated with carbon cycle, climate sensitivity, warming trends, and historical ERF changes, in comparison with their proxy assessment ranges. The consistency between modeled and proxy ranges can be most distinctively shown for warming trends by GMST and OHC changes (Fig. 11 (k) and (l)), with 'Prior' ranges substantially wider and higher than assessed ranges but comparable 'Constrained' ranges. The consistency of sensitivity indicators, including TCRE, (Fig. 11 (h)–(j)) is complex because the proxy assessment ranges (black error bars) themselves are not necessarily consistent with each other, as discussed in the next section, and narrowed from the AR5-assessed ranges (grey error bars). Overall, consistency is better for 'Prior' ranges, although 'Constrained' ranges, considerably narrowed and lowered, are still within the AR5-assessed ranges. The ranges of the carbon cycle indicators (Fig. 11 (a)–(g)), including accumulated carbon and implied cumulative emissions in the historical period 1750–2011, are not significantly different between the two ensembles and broadly agree with assessed ranges. Ensemble runs for the extended historical period starting from 1750 were conducted for calibration. The ranges of the ERF indicators (Fig. 11 (m)–(t)) are consistent with assessed ranges, except for 'Prior' $CO_2$ and 'Constrained' Montreal gases, as mentioned above. Other minor changes from 'Prior' to 'Constrained include a reduced range for aerosols and lowered ranges for stratospheric and tropospheric ozone.



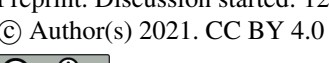

**Figure 11: Distributions of climate indicators: (a) accumulated carbon over ocean at doubling CO₂ in 1pctCO2; (b) same as (a) but over land; (c) same as (a) but at quadrupling CO₂; (d) same as (c) but over land; (e) accumulated carbon over ocean in 1750–2011; (f) same as (e) but over land; (g) implied cumulative emissions in 1750–2011; (h) equilibrium climate sensitivity diagnosed with**
**CO₂ quadrupling forcing (ECS$_G$); (i) transient climate response (TCR); (j) transient climate response to 1000 GtC cumulative CO₂ emissions (TCRE); (k) global mean surface temperature (GMST, air-ocean blended) in 2000–2019 relative to 1961–1990; (l) ocean heat content (OHC) in 2018 relative to 1971; (m) effective radiative forcing (ERF) of CO₂ in 2014 relative to 1750; (n) ERF of CH₄ in 2011 relative to 1750; (o) same as (n) but of N₂O; (p) same as (n) but of 'Montreal gases' (CFCs/HCFCs/halons); (q) same as (n) but of 'F-gases' (HFCs/PFCs/SF₆); (r) same as (m) but of aerosols; (s) same as (n) but of stratospheric O₃; (t) same as (n) but of**
**tropospheric O₃. Error bars and pairs of triangle markers indicate *likely* ranges (17–83%) and *very likely* ranges (5–95%), respectively. The black and grey error bars indicate proxy assessment ranges and AR5-assessed ranges, respectively. The proxy ranges are based on 5–95% ranges of the CMIP Earth system models in (a)–(d), but otherwise taken from the RCMIP Phase 2 protocol that partly includes the AR5-assessed ranges.**



### 3.3 Results: projected warming

Figure 12 illustrates temperature response in two SSP scenarios, SSP1-2.6 and SSP2-4.5, where warming is measured by an increase in global mean surface air temperature (GSAT) relative to 1850–1900, and the period up to 2100 is presented. In the scenario labeled 'SSPn-x.x', 'n' (1–5 numbers) identifies different socioeconomic development pathways, and 'x.x' expresses a nominal forcing level in Wm$^{-2}$ at the end of the 21$^{st}$ century or later. The shaded areas indicate 33–66% ranges. The upper bound corresponds to the level to which warming is *likely* (66–100%) to be limited at the time, while the lower

bound corresponds to the level which warming is *likely* to exceed. These thresholds are shown in Table 1 for peak and end-of-century (end-21C) warming for eight SSP scenarios, where the end-21C period is set to 2081–2100, in accordance with the AR5.

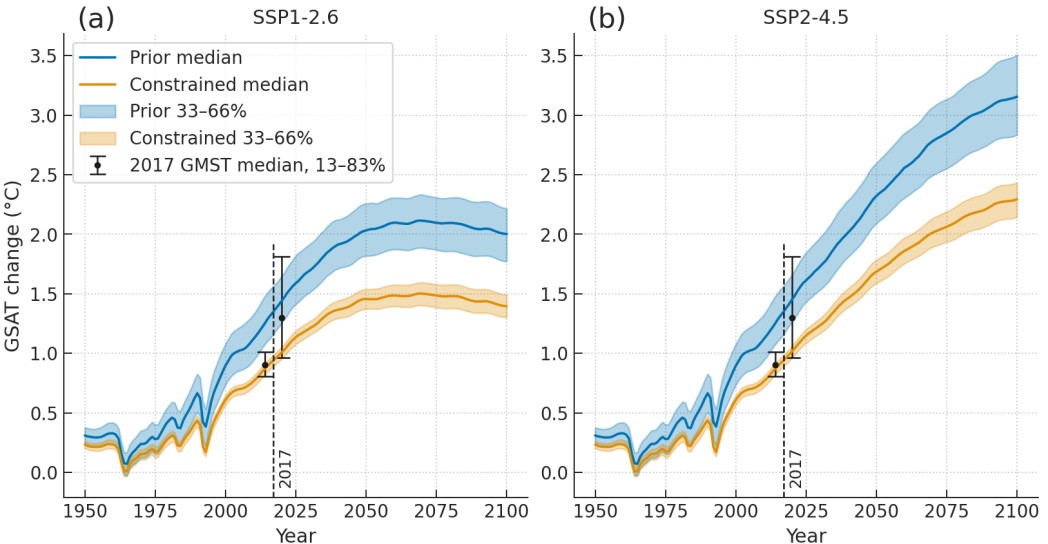

**Figure 12: Global mean surface air temperature (GSAT) changes relative to 1850–1900 in SSP1-2.6 (a) and SSP2-4.5 (b) scenarios**
**from 'Prior' and 'Constrained' ensembles. Medians and 33–66% ranges at each time point are shown by lines and shading. The error bars indicate medians and *likely* (17–83%) ranges of global mean surface temperature (GMST, air-ocean blended) changes in 2017.**

**Table 1: Critical global mean surface air temperature (GSAT) change relative to 1850–1900 in different Shared Socioeconomic**
**Pathway (SSP) scenarios. Warming levels at peak during the 21$^{st}$ century and averaged over the period 2081–2100 (end-21C) are shown for those *likely* to be limited (66 percentile) and *likely* to exceed (33 percentile) from 'Prior' and 'Constrained' ensembles.**

|  | SSP1-1.9 | SSP1-2.6 | SSP4-3.4 | SSP5-3.4* | SSP2-4.5 | SSP4-6.0 | SSP3-7.0 | SSP5-8.5 |
|---|---|---|---|---|---|---|---|---|
| *Likely* limited to at peak | 2.08 | 2.34 | 2.34 | 3.10 | 3.51 | 4.25 | 5.20 | 6.15 |
|  | *1.39* | *1.60* | *2.09* | *2.16* | *2.43* | *2.96* | *3.70* | *4.44* |
| *Likely* limited to at end-21C | 1.82 | 2.27 | 3.05 | 2.78 | 3.38 | 4.01 | 4.72 | 5.54 |
|  | *1.20* | *1.54* | *2.07* | *1.90* | *2.36* | *2.82* | *3.34* | *3.98* |



| | | | | | | | |
|---|---|---|---|---|---|---|---|
| *Likely* exceed at peak | 1.68 | 1.89 | 2.50 | 2.54 | 2.83 | 3.47 | 4.30 | 5.17 |
| | *1.24* | *1.41* | *1.84* | *1.92* | *2.14* | *2.61* | *3.25* | *3.94* |
| *Likely* exceed at end-21C | 1.44 | 1.82 | 2.47 | 2.23 | 2.76 | 3.29 | 3.86 | 4.64 |
| | *1.04* | *1.34* | *1.83* | *1.65* | *2.08* | *2.48* | *2.94* | *3.55* |

Units: °C; * Overshoot type pathway; Upper: 'Prior' ensemble; Lower in italic: 'Constrained' ensemble

Regarding consistency with target warming levels, such as two degrees above preindustrial levels, the 'Constrained'
ensemble agrees relatively well with the AR5 assessment (Collins et al., 2013) for each of the comparable Representative
Concentration Pathways (RCPs, van Vuuren et al., 2011) such as RCP2.6 with SSP1-2.6. For example, AR5 states that end-
21C temperature change above 2 °C is *unlikely* (0–33%) under RCP2.6, which implies that temperature is *likely* limited to
2 °C. This assessment is consistent with the SSP1-2.6 result from 'Constrained' (*likely* limited to 1.51 °C) but not from
'Prior' (*likely* limited to 2.27 °C). Some threshold temperatures in 'Constrained' are not consistent with AR5, such that the
temperature in SSP2-4.5 *likely* exceeds 2.06 °C, while in AR5 it is *more likely than not* (> 50–100%) to exceed 2 °C in
RCP4.5. There is a similar difference in the possibility of limiting to 4 °C in SSP5-8.5 and RCP8.5. AR5 assessed these
cases with *medium confidence* rather than *high confidence*, implying that the reduced *likely* ranges (as in 'Constrained') can
update the AR5 assessment more authentically. However, at present, the 'Constrained' ensemble does not incorporate
possible uncertainties, as discussed in the next section. It should also be noted that SSP forcing is not exactly the same as
corresponding RCP forcing, leading to noticeable temperature differences between the comparable scenarios (Nicholls et al.,
2020).

There are also some issues with handling of historical warming. The AR5 refers to a specific level of 0.61 °C from
HadCRUT data for the period 1986–2005, which is added to the CMIP5 projected warming. However, HadCRUT warming
is defined as an air-ocean blended temperature and is thereby somewhat underestimated for the GSAT definition (Schurer et
al., 2018) with which modeled future warming is evaluated. In any case, the AR5 assessment is effectively constrained by
observed warming, which may be responsible for its better agreement with the 'Constrained' ensemble. Figure 12 indicates
medians and *likely* (17–83%) ranges of temperature changes in 2017 by the GMST (air-ocean blended) definition, 1.30
[0.96–1.81] °C in 'Prior' and 0.90 [0.80–1.01] °C in 'Constrained'. The latter warming levels are also closer to the SR15
assessment of 1.0 [0.8–1.2] °C for human-induced warming (Allen et al., 2018), despite some bias towards the lower end of
the assessed range.



## 4 Discussion

### 4.1 Performance as an emulator

It has already been confirmed that the MCE reproduces many different CMIP models effectively in terms of thermal response to idealized $CO_2$ forcing changes, as demonstrated in Nicholls et al. (2020). The forcing and response parameters

are adjusted to emulate two of the CMIP's basic experiments for step-shaped (abrupt-4xCO2) and ramp-shaped (1pctCO2) forcing increases. The forcing scheme uses different functions depending on concentration levels: a quadratic expression in terms of logarithmic concentrations in the range from two to four times the base level, smoothly connecting to linear expressions outside this range. This flexibility suits the CMIP models' tendency to deviate from logarithmic concentration proportions at higher concentrations, leading to better emulation not only for responses to quadrupling increases commonly

used in basic experiments, but also for responses to considerably lower increases in many mitigation scenarios.

However, the scheme assumes constancy of the climate feedback parameter; emulation accuracy will therefore be decreased in scenarios where state dependency of feedbacks emerges. A typical example appears in a cooling scenario. The RCMIP Phase 1 results include a case in which the MCE fails to emulate a halving $CO_2$ experiment, while successfully emulating both doubling and quadrupling (See Figure 2 of Nicholls et al., 2020). It is also known that state dependency becomes

significant when the time integration of the step response continues over multi-centennial to millennial timescales (Knutti and Rugenstein, 2015; Rohrschneider et al., 2019). As CMIP models tend to deviate from linearity between the TOA energy imbalance and the surface temperature anomaly so that additional warming occurs with time, the MCE would result in underestimated warming in such a case. In practice, this issue is not significant up to the time horizon of 2100, commonly used in mitigation scenarios, in particular for lower than doubling $CO_2$ levels.

For non-$CO_2$ forcing, additivity is assumed across different agents, except for overwrapping effects for $CH_4$ and $N_2O$, as parameterized in Etminan et al. (2016). The forcing amplification for $CO_2$ is not extended to total forcing. These are reasonable assumptions for most mitigation scenarios where non-$CO_2$ components are presumably not extreme.

The carbon cycle module has a mixture of fixed and adjustable parameters, including those for several feedback mechanisms from temperature changes. The current configuration successfully works to represent the CMIP ESMs' ranges in terms of

carbon budget in the idealized 1 %-per-year $CO_2$ increase experiment. However, it has not yet been verified that each of the ESMs can be accurately emulated.

Diagnosing the carbon cycle parameters to individual ESMs is a main issue to be addressed in future. Accumulated carbon in response to atmospheric $CO_2$ input has a trade-off relationship between ocean and land, and both components have their own mechanisms of climate-carbon cycle feedbacks, which are also subject to the magnitude of temperature response. This

implies that calibrating the MCE parameters for each ESM requires a series of pulse-response experiments designed to allow each of the ocean and land contributions to be isolated, with and without temperature feedback. Besides the standard 1%-per-year increase experiment, the CMIP6 provides idealized ESM experiments, including 1%-per-year increase variants with different configurations and a variety of pathways to zero emissions (Jones et al., 2019; Keller et al., 2018). The extent to





which different ESMs are emulated for these scenarios needs to be verified with calibrated parameters, leading to further
insights into carbon-cycle behavior in terms of amount of emissions, hysteresis effects after attaining zero emissions, and
state dependency.

While the covariance of MCE parameters is incorporated for the CMIP models' variability of $CO_2$ induced warming, the
carbon cycle parameters and the non-$CO_2$ scaling factors are independently sampled. There may exist other covariance
between key indicators, such as a correlation between $CO_2$-induced warming and aerosol cooling implied from CMIP6
historical experiments (Meehl et al., 2020). As different types of aerosol schemes constitute a major source of model
variations, incorporating covariance associated with aerosol forcing would improve parameter sampling, leading to more
appropriate indicator ranges.

The results shown in the previous section are outputs from concentration-driven experiments, where implied emissions are
available for $CO_2$ only. Likewise, the emission-driven option is currently limited to $CO_2$. The two types of experiments are
equivalent within numerical errors associated with a time integration scheme, for which Runge-Kutta 4th order is used.
However, implied emissions tend to be noisy when pulse-like non-$CO_2$ forcing is given, owing to the temperature
dependency implemented in carbon cycle modules. This is the case in historical experiments including volcanic forcing.

## 4.2 Further improvement on constraints

The 'Constrained' ensemble was applied to that compared in the RCMIP Phase 2 exercise, where the MCE is recognized as
one of two models that match relatively well the target constraints, among nine participant models with different degrees of
complexity (Nicholls et al., personal communication). The MCE is a relatively simple emulator, conceivably cited with a
simple thermal response, an intermediate-complexity carbon cycle, simply parameterized non-$CO_2$ GHG forcing, and no
other Earth system components. This simplicity and the successful results obtained imply that a method with less
complicated structures and fewer control parameters offers advantages when building reasonable parameter ensembles,
despite less capacity to emulate detailed Earth system components.

Several issues require clarification with regard to the differences between 'Prior' and 'Constrained' ensembles.

It should be emphasized that the constraints used in the RCMIP are preliminary, wherein the formal IPCC Sixth assessment
is not yet available (Nicholls et al., personal communication). The current proxy constraints such as the ones for the three
climate sensitivity ranges of $ECS_G$, TCR, and TCRE adopted from individual studies are not necessarily consistent with each
other. The range of $ECS_G$ is based on multiple lines of evidence, including feedback process understanding, historical
records, and paleoclimate records (Sherwood et al., 2020). Here, $ECS_G$, rather than ECS, is referred to, assuming that process
understanding is largely based on the CMIP's quadrupling $CO_2$ experiments. The range of TCR is based on 30 and 22
AOGCMs from the CMIP5 and CMIP6, both constrained by warming trends during recent decades (Tokarska et al., 2020).
In contrast to these observations and modeling studies, the range of TCRE is based on 11 CMIP6 ESMs (Arora et al., 2020).
Improved ensembles based on comprehensively assessed constraints would increase reliability of probabilistic projections,
leading to better insights into future warming.





In comparison with the AR5 assessment, the 'Constrained' ensemble has considerably low-biased climate sensitivity, but nevertheless indicates comparable future warming across different scenarios. As stated above, this inconsistency can be partly explained on the basis of the AR5 method for warming levels that adds up observed historical warming to CMIP5-modeled future projections. With regard to consistency throughout the whole period, the emulator approach would be more desirable. In any case, it is necessary to impose appropriate weighting on CMIP models to be emulated, in particular, when the model ensemble has a wide spread and some outliers in terms of reproducibility of past and current climates (Cox et al., 2018; Tokarska et al., 2020). The MH sampler with observed warming constraints corresponds to an indirect method for such weighting. As the present results decisively depend on surface temperature and OHC observations during recent decades, their validity as a constraint needs to be discussed from a broad perspective across forcing, response, and sensitivity. The current constraining assumes observed warming as entirely forced response. Recent findings from warming attribution studies may support this, suggesting that human-induced warming is similar to observed warming (Allen et al., 2018). However, the attribution depends on temporal and spatial patterns of forced response in multiple AOGCMs as well as their diagnosed forcing, leading to a complicated situation in which constraining data and AOGCMs to be constrained are mutually dependent. Besides, there remain substantial uncertainties of response patterns to changes in individual forcing factors owing to the diversity of AOGCMs (Jones et al., 2016). Moreover, the new CMIP6 models appear to have marked differences in the magnitude of internal variability underlying attribution studies (Parsons et al., 2020). The GMST constraint does not consider such uncertainties and may be replaced with broader ranges from new insights into forced response.

Furthermore, it is also necessary to constrain forced response on a centennial timescale. Besides the HadCRUT data available since 1850, the OHC data, limited to the late $20^{th}$ century, may include delayed response components on a much longer timescale. In fact, major volcanic eruptions occurred frequently in the $18^{th}$ and $19^{th}$ centuries. The initial year of 1850 is commonly used as a proxy preindustrial time point to avoid difficulties that arise from limited observations and major eruptions (Allen et al., 2018). However, the pre-1850 volcanic impact on the deep ocean should be examined carefully. In fact, it has been recognized in historical runs with the MCE that the OHC increase during the late $20^{th}$ century significantly depends on the initial year, while the surface warming does not. A series of volcanic eruptions in the period 1750–1850 appears to contribute to heating after the period and amplifies heat content increase since 1850. The results suggest that human-induced OHC increase should be distinguished from observed total increase for better constraints.

Aerosol cooling is one of the key factors influencing temperature changes in the latter half of the $20^{th}$ century, resulting in different ranges of other climate indicators. The present method relies on the prescribed ERF time series prepared for the CMIP6, which is scaled to the proxy assessment range in 2014 (Fig. 11 (r)). Although this procedure assures that cooling magnitude is constrained to the given range at the specific time point, the base time series is fixed. The constraint, adopted from Smith et al. (2020), is the outcome of the Radiative Forcing Model Intercomparison Project (RFMIP, Pincus et al., 2016), one of the CMIP6-endorsed model intercomparisons. A better insight into aerosol forcing may update its historical time series, thereby leading to improvement of forced response and other climate indicators, including climate sensitivity.



Technical issues exist when sampling from the 'Prior' ensemble with observed warming constraints, associated with their distinct differences. The proxy ranges of the constraints are much more biased and localized compared to the 'Prior' distributions, leading to inefficient sampling. The acceptance rate in the present case reached only about 0.6 %, requiring over 100 k-member calibration runs to obtain a 600-member 'Constrained' ensemble. Besides the efficiency issue, the

sampling process should be visually monitored to verify whether the acceptance/rejection is reasonable. As the MH independence sampler compares a probability density ratio of the next state to the current between candidate and target densities, care is to be exercised at the distributions' tail regions where relatively large approximation errors may exist. The present method introduced ad hoc criteria to avoid acceptance with an unexpectedly large density ratio.

## 5 Conclusions

A new climate model emulator, MCE, was developed, and its probabilistic climate projections for representative scenarios were demonstrated and thoroughly discussed. The MCE is based on impulse response functions and several parameterized physics including key climate-carbon cycle feedbacks and may be categorized as a relatively low complexity model among recent model intercomparison participants. It has an advantage when building reasonable perturbed ensembles transparently, despite its lower capacity to emulate detailed Earth system components. Perturbed ensembles can cover complex climate

models' diversity effectively, reflecting their covariance structure of diagnosed forcing-response parameters associated with $CO_2$-induced warming. Probabilistic projections constrained with several ranges of climate indicators, including $CO_2$ and other forcing factors and observed warming trends over recent decades, suggest that complex climate models generally overestimate climate sensitivity. The sampling procedure implemented for parameter constraining, based on a Metropolis-Hastings algorithm, effectively works as weighting given to complex models.

Results from climate assessments for future scenarios in terms of their compatibility with climate mitigation goals are preliminary, and experiments should be conducted with newly assessed constraining data. There are considerable uncertainties about forced components of historical warming as well as different forcing factors, and consistency of assessed ranges among different climate sensitivity metrics. These are main issues to be clarified in the forthcoming new assessment. There is some room for improvement in emulator functionality. The carbon cycle module has not been configured to

individual complex models, full emissions-driven experiments have not been supported, and perturbed parameter ensembles have not reflected full covariance structures of complex models. These issues are to be addressed in future work.

*Code and data availability.* MCE source code and example usage scripts as used in this submission are available from the MCE GitHub repository at https://github.com/tsutsui1872/mce (last access: 16 March 2021) and archived by Zenodo

(https://doi.org/10.5281/zenodo.4604695, Tsutsui, 2021).

*Author contribution.* JT developed the model code, performed experiments, and prepared the manuscript.



*Competing interests.* The author declares that the author has no conflict of interest.


*Acknowledgements.* I acknowledge the steering committee and the members of RCMIP, in particular Zebedee R. J. Nicholls, for giving me the opportunity to participate in the model intercomparison. The scenario experiments with perturbed parameter ensembles in my study depend on input scenario data and constraining data provided from RCMIP.

*Financial support.* This work was supported by the Integrated Research Program for Advancing Climate Models (TOUGOU) Grant Number JPMXD0717935457 from the Ministry of Education, Culture, Sports, Science and Technology (MEXT), Japan.

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
