# Peer review of "Minimal CMIP Emulator (MCE v1.2): A new simplified method for probabilistic climate projections"

_Geoscientific Model Development, 2021_

## Author Comment (AC3)

**Response to referee comments 1**

This study describes the Minimal CMIP emulator (MCE) model for projections of global mean surface air temperature and presents probabilistic ensembles generated using a Monte Carlo approach.

In the study, a set of 600 simulations is generated to construct a prior ensemble, with model parameters varied according to estimates of the relevant properties in the literature. A constrained ensemble is then also generated using observational constraints to train an additional 600-member ensemble.

This approach has been well justified within the manuscript, and links well to existing current literature (such as the RCMIP of other computationally fast climate models).

I find the study well written, and I am confident that both the model and Bayesian approach to ensemble generation will be a useful addition to the literature. I find the current version of the manuscript is close to recommendation for publication. I do have some comments and concerns that I would like to see addressed in a revised version and/or explained in a response, before recommending publication of the manuscript. I outline these comments and concerns below.

Comments/concerns:

**(1) Schematic of the MCE**

The description of the model (Section 2) is well written, and I came away understanding how the model was constructed mathematically. There is enough information there for someone to code their own version of this model if required.

However, there were instances where I think a schematic of the MCE, or some other representation that would show how the equations are applied, would be very helpful to the reader.

For example, a schematic showing exactly what is meant by the 'composite atmosphere-ocean mixed layer' in equations (3) to (6) would be helpful to more quickly understand how carbon in the atmosphere and surface ocean mixed layer are being treated.

One reason I find that a schematic would help the reader tremendously is that I couldn't work out whether the MCE uses different representations for heat and for carbon, or the same representations. For example, Lines 299 to 302 indicate that temperature is effectively treated as a three layer heat-exchange model:

"The IRM for temperature change is transformed into a three-layer heat exchange model in physical space. When diagnosing the CO2 forcing and response parameters, the top layer temperature was treated as global mean surface air temperature (GSAT). As in HadCRUT GMST was defined as a surface air ocean blended temperature change; here, a factor of 1.04 was used to convert observed GMST change into the MCE's GSAT change."

A schematic would help the reader work out whether the top layer in this heat exchange is the same thing as the composite-atmosphere-ocean mixed layer' for carbon. At present I am unsure, and so I cannot see whether observational records of Global Surface Air Temperature, Global Surface Temperature or Global Sea Surface Temperature would be the best type of record to compare this top layer with (e.g. if the top thermal layer is meant to represent the heat uptake by the composite surface mixed layer and atmosphere {as in the carbon-component of the model}, then an SST record is the best one to compare it to since the surface mixed layer is the majority of the heat uptake. However, if the top layer is meant to represent the temperature response of the atmosphere only, then Global mean Surface Air Temperature records are best, as currently used).

In the first paragraph of Subsection 2.1, I have added new Figure 1, as well as introducing text, illustrating the MCE's components in a box-model form. Since each component is formulated on its own impulse response functions, the boxes are separately represented for the thermal response and carbon cycle modules. For the composite layer of the carbon cycle module, I have added a reference to this figure in the text where this term is first mentioned (L114 in the new revision, hereinafter). For the top layer of the thermal response module, I have added a sentence to describe its representative Earth

system components (L326). The top layer is conceptually treated as a fast-responding component, and in practice its temperature is calibrated with the global mean surface air temperature from AOGCM output.

**(2) Development of 'Constrained' ensemble.**

I would like to see the tests used for the derivation of the 'constrained' ensemble more explicitly stated. At present the manuscript reads (starting line 292):

"The 'Constrained' uncertain parameters were sampled from those of 'Prior' through a sequence of the MH sampler with a subset of RCMIP constraints, as follows: (1) CO2 ERF in 2014 relative to 1750 evaluated in Smith et al. (2020), (2) TCR estimated in Tokarska et al. (2020, Table S3, both constrained), and (3) GMST in the period 1961–1990 relative to the period 2000–2019 from the HadCRUT.4.6.0.0 dataset (Morice et al., 2012) and ocean heat content (OHC) in 2018 relative to 1971 from the dataset of von Schuckmann et al. (2020). In this case, in addition to 1pctCO2 runs, historical scenario runs with a set of proposed parameters were conducted to obtain data fed into the sampler."

It would help the reader reproduce an equivalent constrained ensemble, perhaps with different simple climate model, if these 3 tests were explicitly stated. Possibly a table could be used, or else mathematical functions for the tests given, in an Appendix or supplementary material?

**I have added a new Jupyter notebook in the code repository to fully describe the derivation of the 'Constrained' ensemble, available at**

https://github.com/tsutsui1872/mce/blob/master/notebook/t\_genparms\_rcmip2.ipynb. This electronic notebook contains program code to reproduce the ensemble step by step, including figures and tables explaining input materials and the results of each step. I have released MCE v1.2.1, a minor revision including the notebook, and have revised *Code and data availability* section as well as reference to the code repository in the text (L348) accordingly.

**(3) Relevant literature for context.**

The placing of the manuscript within existing literature is a general strength of the study. For example, the comparison to the 600-member ensembles produced by the MAGICC emulator is presented well, as is the comparison to the FaIR model, allowing the reader to understand the key similarities and differences between the approaches of the MCE model used here and these previous models and approaches. Also, the placing of the MCE within the RCMIP provides the reader with necessary context.

However, the manuscript could be improved further by placing the work within context of at least one recent climate model study that has also adopted a similar Bayesian approach to ensemble generation, for example the recent studies by Goodwin and Cael (2021) and Goodwin (2021).

The methods used here and in Goodwin and Cael (2021) and Goodwin (2021) adopt a Bayesian approach to ensemble generation, with both the similarities and differences between them. For example:

- (i) Goodwin and Cael (2021) use a much larger prior ensemble with an observational filter used to either accept or reject each simulation to form the posterior, whereas this study uses a sampling algorithm to keep the prior sample size smaller.
- (ii) Goodwin and Cael (2021) vary a set of ~25 model parameters independently (including model parameters for radiative forcing) in the prior ensemble, and then allow any relationships between variables to emerge within the posterior ensemble. These relationships are then explored with a principle component analysis of the parameters within the posterior ensemble. In contrast, this study varies the model parameters using pre-defined relationships between varied parameters in the prior, using principle components to evaluate what those pre-defined relationships should be.
- (iii) Both studies use observational records of surface warming and heat uptake to constrain the ensembles, but with a different approach.

Placing this study's method in context with previous Bayesian approaches to climate model ensemble generation within the literature would be helpful to the reader. Perhaps some of the findings from the Principle Component and other analyses in Goodwin and Cael are consistent with some of the findings of this study?

I have added two paragraphs in Subsection 2.5 after its leading paragraph to place the current method in context with other similar methods, as the reviewer suggested. Considering technical details added in response to the second comment, I have mentioned an issue of the MH algorithm implementation in the new paragraphs, and have made some revisions to relevant parts: one new paragraph of Subsection 3.1 at L343, and the last paragraph of subection 4.2.

**(4) Line 135: Temperature IRM split into three time-constants:**

"The IRM of the temperature change defines three components with typical time constants of approximately 1, 10, and > 100 years. Although the temperature response is usually well represented by two separated time constants of approximately 4 and > 100 years (e.g., Held et al., 2010, Geoffroy et al., 2013), ..."

There is an additional reason why having the three time-constants may be important, and be of benefit to the MCE model: The sea surface warming pattern effect. The climate feedback and warming responses of complex CMIPclass models to radiative forcing are found to be affected on a (multi-)decadal timescale by the changing pattern of surface warming (e.g. see Andrews et al., 2015). It seems to me that to emulate the impact of this pattern effect on surface warming in a CMIP-class model using MCE it is a benefit to have three timescales, and that this should be highlighted in the study.

**I have added a sentence (L148) to describe an additional benefit of using three time constants, as the reviewer suggested.**

Minor points:

Abstract, line 17:

"... CMIP- and observation-consistent ensembles ..."

At present, the hyphen is critical to the meaning of this sentence: if the hyphen were not there then you would be comparing the CMIP simulations to the observation-consistent simulations — which is how I originally read it. Actually, the author is comparing the CMIP-consistent simulations to the observation-consistent simulations.

To avoid confusion, here I suggest changing this to "... CMIP-consistent and observation-consistent ensembles".

**This has been corrected as suggested.**

Line 315:

"Other details of experimental specifications are provided in the MCE's code repository."

It would be helpful to put the reference or doi of the code repository here.

**I have added a reference to the section *Code and data availability* at the end.**

References:

Andrews, T., Gregory, J.M., and Webb, M.J. (2015) The dependence of radiative forcing and feedback on evolving patterns of surface temperature change in climate models. J Clim 28, 1630–1648. https://doi.org/10.1175/JCLI-D-14-00545.1.

Goodwin, P. and Cael, B.B. (2021) Bayesian estimation of Earth's climate sensitivity and transient climate response from observational warming and heat content datasets, Earth System Dynamics, 12, 709–723, 2021, https://doi.org/10.5194/esd-12-709-2021

Goodwin, P. (2021) Probabilistic projections of future warming and climate sensitivity trajectories, Oxford Open Climate Change, kgab007, https://doi.org/10.1093/oxfclm/kgab007

**Response to referee comments 2**

This paper uses a Bayesian method to produce constrained projections of future warming in a three time-constant impulse response climate model (the Minimal CMIP Emulator, MCE v1.2). I find the level of detail given to be sufficient and the model description technically sound. The carbon cycle dynamics are impressive. The model is run in configurations that both emulate CMIP6 model behavior (blue lines and in section 3) and a "constrained" ensemble set (orange) that is constrained to ranges of key climate variables that are discussed in Nicholls et al. (2021).

My only major comment is not on the model itself but in the example application. The low projections of the constrained ECS and warming — assuming the ranges of Nicholls et al. (2021) which used the Sherwood et al. (2020) ECS assessment — are possibly too low. Sherwood et al. (2020) gives an ECS very likely (interpreted as 90% probability) range of 2.3–4.7 K, whereas the constrained distribution (I'm reading off figure 8 d) has a 90% range of less than 3.5 K. The implications of this are seen clearly in Figure 12: compare AR6 projections in Chapter 4 of that report (for SSP1-2.6, which is a "well-below 2C" rather than a "1.5C consistent" scenario) and assessment of present-day temperature in Chapter 2. The "constrained" range seems over-constrained (compare projections for SSP1-2.6 in AR6). Can you really have this much confidence in the future warming? I would just like some additional explanation as to why the model seems to provide such low constrained estimates of future warming (and whether alternative constraints would show something different). Such a discussion is important, if the aim for this model is to be used by the climate change mitigation community.

I have revised the second paragraph (the second and third paragraphs in the first revision) of Subsection 4.2 to draw further attention to the concerns the reviewer pointed out.

Indeed, the 'Constrained' ensemble projects low warming with narrow uncertainty ranges, compared to the AR6 projections. This is simply resulted from constraining with the proxy assessed ranges, in particular, recent warming trends adopted from specific datasets. I am confident about the results in the RCMIP2 framework, but not necessarily about their reliability as possible future projections.

As already discussed, the RCMIP2 proxy assessed ranges may not be consistent with each other. Now, I can explore this issue quantitatively.

In the AR6 assessment, the human induced warming in 2010–2019 relative to 1850–1900 is 1.07°C with a *likely* range of 0.8 to 1.3°C in GSAT (3.3.1.1, Table 3.1), and the Earth's total energy gain over the period 1971–2018 is 434.9 ZJ with a *very likely* range of 324.5 to 545.5 ZJ (7.2.2.2, Table 7.1). In the RCMIP2 proxy assessed ranges, surface warming from 1961–1990 to 2000–2019 is set to about 0.54°C with a *very likely* range of about 0.46 to 0.61°C in GMST, and ocean heat content change from 1971 to 2018 is set to 320.7 ZJ with a *likely* range of 303.7 to 337.7 ZJ. Although the two sets of assessed ranges can not directly be compared, the RCMIP2 numbers imply smaller trends and narrower uncertainty ranges than the AR6 numbers. This is confirmed from an experimental ensemble constrained with the AR6 ranges, which I am now conducting.

As to the current manuscript, I need to avoid referring to AR6 knowledge for consistency as I conducted this study and prepared the manuscript well before the AR6 publication. I will report new findings based on the AR6 assessed ranges in a separate paper.

Specific minor comments:

Line 36: more specifically, the Working Group 1 contribution

**This has been added.**

Line 38: insert "the" before "climate assessment"

**This has been corrected as indicated.**

Line 50: reduced from > smaller than

**This has been corrected as indicated.**

Line 51: RWF: state that RWF is TCR / ECS

I think that the TCR-to-ECS ratio is a specific case of the realized warming fraction that depends on forcing pathways and elapsed time. I have made a slight revision and referred to the sensitivity ratio in this sense.

Line 54–55: on aerosol forcing in CMIP5 and CMIP6 — the aerosol forcing is part of the story but not all; as a shameless plug see

Smith and Forster (2021): https://agupubs.onlinelibrary.wiley.com/doi/10.1029/2021GL094948

**This has been revised, as suggested, to state GHG forcing differences as another possible cause with the new literature.**

Line 62: Nicholls et al. (submitted) is now published in Earth's Future (and line 279).

**These have been refreshed. Also, "Nicholls et al., personal communication" in Subsection 4.2 has been revised considering the published contents.**

Line 109: "about" 0.2 : would it not be asymptotically exactly 0.2, as this is the partition fraction of the infinite time constant box. I'd just suggest removing "about".

Although the asymptotic long-term airborne fraction is close to 0.2 for a small pulse input, it becomes greater with the pulse size due the buffering mechanism of sea water. The value is about 0.204 for the 100 GtC input. This mechanism is responsible for the nonlinear relationship between the DIC and  $CO_2$  concentration shown in Figure 5(a). A new sentence has been added to explain the approximate value.

Line 125–126: the text describes 17th and 83rd percentiles but the caption and legend in figure 3 say 5 and 95.

**This was my error and has been corrected.**

Line 138: "such as volcanic eruptions and geoengineering mitigation" — these results are also mentioned in Leach et al. (2021: https://gmd.copernicus.org/articles/14/3007/2021/gmd-14-3007-2021.pdf) and Cummins et al. (2020: https://journals.ametsoc.org/view/journals/clim/33/18/jcliD190589.xml) that advanced the benefits of three-box IRMs — might be worth mentioning.

**I have added two references with a few words. Referring to Leach et al. (2021) could make a circular referencing. Instead, I have referred to Gupta and Marchall (2018) in the volcanic forcing context.**

Line 228: AR6, WG1, Chapter 7 supplementary material updates this.

**This is true, but I am sticking to AR5 because of consistency with the configuration in RCMIP2, and have rephrased "currently" to "at the time of this manuscript preparation."**

Lines 229–231: that's a shame, but OK. No need to change the model, but it's not too difficult to include these elements. The non-CO2 components could be borrowed from FaIR (Leach et al. 2021), and uncertainties for aerosol forcing in CMIP6 models can be obtained from 11 models that participated in RFMIP and AerChemMIP historical aerosol forcing experiments — tuning the parameters similarly to what has been done in this paper for the carbon cycle with the C4MIP models in Arora et al. (2020) (see Smith et al. 2021, https://doi.org/10.1029/2020JD033622).

I appreciate the reviewer's suggestion. Since AR6 publication, I have been intensively working on implementing full forcing schemes and functions of AR6-consistent emission-driven ensemble runs. I will report these new components separately.

Line 330: 90% and 66% of members

**This was my error and has been corrected.**

Line 352: "...and the 'Constrained' trend appears underestimated." Some variation in the forcing time histories, particularly for aerosols, could help here. See Smith et al. 2021: https://doi.org/10.1029/2020JD033622

**I have added a new sentence with the literature to state historical aerosol forcing variations as the reviewer suggested.**

Line 403: "or later" is incorrect: the SSP forcing label refers to 2100.

**I had intended to refer to a nominal stabilizing level assumed in the original RCP, but this has been deleted.**

Fig 12a, legend: 17-83%?

The ensemble runs are shown with 33–66% ranges to focus on 66-percentile levels from the lower and upper ends, as in Table 1. The 2017 GMST is indicated with 13–83% ranges (*likely* ranges) for comparison with SR15-assessed human-induced warming.

Fig 12: I can't determine what the black error bars are showing — if it is GMST from observations, why they are different for blue and orange distributions, and if they are from the ensembles, why they don't exactly overlap the uncertainty plumes (in which case they would be superfluous anyway).

The black error bars are shown as a reference for comparison with the SR15-assessed GMST warming. The MCE results are shown by GSAT changes relative to 1850–1900 for comparison with the AR5assessed future warming levels. In the current configuration, a factor of 1.04 is used to convert GMST to GSAT although the relationship between the two metrics is too complex to be related proportionally. The vertical differences between the black dots and the location of the blue and orange lines are due to this conversion factor.

For a visual purpose, the two error bars are slightly shifted from the reference year of 2017 on the horizontal axis. This explanation has been added in the figure legend.

As described in the response to the previous comment, the different extent between the error bars and blue/orange plumes is due to percentile differences.

Lines 446-450: can you clarify whether the CO2 forcing expression is only valid in the 1x to 4x range?

**I have slightly revised Subsection 2.4 to clarify the valid range of a $1 \times$ to $4 \times$ concentration level.**

Line 460: overwrapping > overlapping

**This was my error and has been corrected.**

Lines 479–480: Actually, there is no correlation between the strength of aerosol cooling and CO2 warming in CMIP6. See Smith et al. (2020: https://acp.copernicus.org/articles/20/9591/2020/acp-20-9591-2020.html, figure 9, and I also made this Twitter thread showing the difference compared to the Meehl paper step-by-step: https://twitter.com/chrisroadmap/status/1297798254789263360)

**I have deleted this example as possible covariance between key indicators. From the new literature pointed out by the reviewer, I have realized that a few outlier models made a spurious correlation.**

Line 491: "cited with" probably not the correct word choice here. "Driven by"?

**This sentence is intended to give main features with which the model may be mentioned in some documents.**

Line 517: "human-induced warming is similar..." also found to be true in AR6 (sorry, I know you submitted this paper before AR6 became publicly available, and I was heavily involved the report so I see AR6 statements everywhere).

**This is true, but I am still avoiding referring to AR6 knowledge for consistency because I had to conduct this study without owing to it.**

Lines 529–533: Just a note to say I agree, and this actually was considered in the emulator runs for AR6. A long (2250 year) pre-industrial time series of volcanic forcing was used before the anthropogenic forcing starting in 1750 was assessed.

**I appreciate the comment.**

---

## Author Response (AR2)

**Response to referee comments**

I really appreciate that careful review. I have prepared this revision considering the reviewer's comments as well as the editor's suggestion.

I accept the argument for the lower estimate of warming in this configuration of MCEv1.2 than in say Sherwood et al. (2020) for ECS and AR6 for warming projections is caused by a lower estimate of ocean heat content in RCMIP2. It is curious though that this is having such a large influence on the posterior here. Referring to figure 1 of Nicholls et al. (2021), we can see that MCE indeed has a low projection of ECS compared to other models using this same set of constraints. Anyway, we know that how simple models are each constrained slightly differently and as the author points out, the constraints are inconsistent. As in my previous review I don't suggest running new simulations. However, I do think that the results should be put in context of the warming projections in AR6, even though the study was prepared before then, because it is a landmark publication and the reasons for the differences between your projections and those in AR6 be explained because readers will question this. In fact, the projections are significantly below what would be implied from an AR5 distribution of ECS, also.

I have revised the second paragraph of 4.2 to draw more attention to possible biases in the constrained ensemble run. To put this in context of the warming projections in WGI AR6, I have cited its SPM.

In fact, conforming to given constraints in the RCMIP2 was tricky, and MCE was configured to preferentially match observed surface warming and ocean heat content increase in recent past rather than proxy-assessed climate sensitivity. Although the results may be biased from AR6-assessed indicators, I think the constraining procedure is a good example in this model development paper.

Minor: fig. 13a — just want to bring up a comment from the first review. Is it definitely 13–83%? This is an asymmetric range. 17–83% is not uncommon as is the middle 2/3 and approximately 1 s.d. from a normal — did you mean this?

I have corrected the legend label to 17–83%. I apologize that I didn't properly respond to the reviewer's original indication on this error.

---

## Author Response (AR3)

**Response to editor comments**

Thanks for the edits. Can you also look at the Conclusion in the same context? For example, I am not convinced that the sentence below is supported, especially when we consider how the AR6 has incorporated the outputs of the CMIP MME into its probabilistic projections.

"Probabilistic projections constrained with several ranges of climate indicators, including CO2 and other forcing factors and observed warming trends over recent decades, suggest that complex climate models generally overestimate climate sensitivity"

Clearly there are some higher sensitivity models in the ensemble, which (as I understand it) mostly appear to be producing more unrealistic outputs that the more moderate sensitivity models. But there still exist moderate sensitivity models in the ensemble, so I'm not convinced that "generally overestimate" is a robust statement. In addition, a small biased ensemble is of little value for probabilistic projection, because we do not trust it to be reliable. For this reason, I would personally prefer to start with an ensemble that I believe to be a bit on the wide side.

Thank you so much for your kind suggestion. I have revised the Conclusion and Abstract to more focus on a methodology aspect rather than the differences between the two ensembles. I also took into account the timing of WG1 AR6 publication.

In terms of the ensemble coverage, I agree with the editor's thought. In fact, the MCE method starts with a wide range of model parameters, where CMIP models' diversity is reflected as much as possible, and constrains them to match given indicator ranges if needed. I think this concept is quite natural, and the present method has some advantages to do so, as described in the paper.